# New Developments for the Sustainable Exploitation of Ornamental Stone in Carrara Basin

**Federico Vagnon** [1] , **Giovanna Antonella Dino** [1] , **Gessica Umili** [1] , **Marilena Cardu** [2] and **Anna Maria Ferrero** [1,*]

1   Department of Earth Sciences, University of Turin, 10125 Torino, Italy; federico.vagnon@unito.it (F.V.); giovanna.dino@unito.it (G.A.D.); gessica.umili@unito.it (G.U.)
2   Department of Environment, Land and Infrastructure Engineering, Politecnico di Torino, 10129 Torino, Italy; marilena.cardu@polito.it
*   Correspondence: anna.ferrero@unito.it; Tel.: +39-0116705114

**Abstract:** The use of natural stone has historical and environmental value that makes it strategically valuable for landscape conservation in Europe. Marble, among others, is widely spread on Earth, and it offers high-performance features in architectural applications. However, the complexity of these formations and the rock variability in different ore bodies require detailed studies of the natural and induced stress state, the fracturing degree, and the influence of external factor (such as temperature and/or chemical agents) on the mechanical properties in order to optimize the exploitation processes by reducing extractive waste. This article shows a series of studies conducted by the authors over the last 20 years aimed at making the exploitation of marble blocks in the Carrara basin safer, more efficient, and, therefore, more sustainable. In particular, studies for increasing our knowledge of the natural and the induced stress state through on-site measurements and numerical modeling, studies to improve the quality of the exploited material through improvements of cutting technologies, studies to improve the knowledge of the mechanical behavior of the material under varying loads and temperature conditions, and studies to improve the reuse of water materials and their reduction are reported.

**Keywords:** ornamental stone; rock mass state of stress and fracturing; marble exploitation techniques; waste reduction

## 1. Introduction

According to the Cambridge dictionary, the definition of sustainability is "the idea that goods and services should be produced in ways that do not use resources that cannot be replaced and that do not damage the environment" [1].

Concerning mineral resources, the primary method of resource extraction is still excavation. Since it is impossible to replace them, we must reduce the damage to the environment. To date, it appears that the main constraints to sustainability in the mining sector derive from the consumption of resources needed to extract and process and the increasing contamination generated by the extraction process. According to the United Nations [2], suggestions to maximize the development benefits of mining while improving the environmental and social sustainability of the mining sector were first addressed in the Johannesburg Plan of Implementation (JPOI), where three priority areas were identified, including addressing the environmental, economic, health. and social impacts and benefits of mining throughout their life cycle, including workers' health and safety. The European Union has also addressed research toward these areas by financing important projects in this field (such as "Selective and Sustainable exploitation of Ornamental Stones Based on Demand", SUSTAMINING [3])

devoted to reducing the environmental impacts of stone extraction, mainly through the reduction in the quantity of stone waste and the rational management of stone resources. The application of nondestructive geophysical tests and modelling has been studied in this project to improve our rock mass knowledge.

In this paper, the authors' research and strategies to improve the mining cycle in marble exploitation and to reduce its impact are reported.

*Sustainable Exploitation of Natural Stone: An Overview of the Main Aspects Involved*

The exploitation of natural stones from ancient times to the present proves the historical and cultural relevance of these materials, highlighting their importance in the economy, history, and traditions of the different cultures that have developed over the centuries in Europe. Marble, among others, is a fairly common rock type and offers high-performance features for architectural applications worldwide. However, the complexity of these geological units, on the one hand, and the rock variability in different occurrences, on the other hand, determine the need to study these formations. In particular, it becomes fundamental to classify the geological level of marble, optimize the exploitation process, determine the lithological and deformational features, and, finally, reduce the waste produced during exploitation and processing.

Dimension stone quarrying introduces very peculiar characteristics compared to other extractive industries; it is characterized by an international market, and, above all, it involves high commercial prices on average, which can balance the high production costs typically faced by quarrying companies.

In order to achieve sustainability in this traditional and important activity, the following issues have to be addressed:

- The need for "planned management" and good organization of the activity. In practice, starting from a better knowledge of the stone resources it is essential to plan the use of the land and manage the production/transformation processes adequately.
- The urgency of reducing the generation of quarry waste "at source" through the adoption of the best available exploitation techniques and the introduction of increasingly "precise" technologies. On the other hand, we need to enhance sustainability by a productive utilization of the processing waste.
- The need to guarantee the compatibility and environmental sustainability of the mining activity through an effective evaluation of the quarry planning, the improvement of environmental performance during the activities, and the site's complete rehabilitation at the end.
- The performance depends on the type of cutting machine as well as the rock/rock mass characteristics. Machine specifications are generally easily known, but the rock/rock mass characteristics are not readily available. The rock/rock mass characteristics are of paramount importance, given the fact that the cutting tool directly engages with the rock to be cut. This fact warrants a geotechnical investigation in conjunction with the equipment used in a given site, as there is a direct interaction of the cutting tools with the rock/rock mass which they are used to cut. Moreover, extraction also requires planning ahead where and how to cut in order to minimize waste.
- To improve the knowledge of the behavior of both the rock mass and the rock material.

An unavoidable starting point for improving marble mining is a deep knowledge of the deposit and the territory in which it is located. In fact, many aspects concerning the physical and mechanical characterization of marble and its geological and mining properties are not known sufficiently well.

The exploitation of ornamental stone and marble requires the extraction of intact blocks that can be sawed into slabs and tiles. Therefore, the need to extract intact blocks leads to requirements different from those for, e.g., the extraction of aggregates, and has led to the development of specific exploration and extraction techniques.

The state of natural acting stress and fracturing of the rock masses is the primary point by which to define the possibility of extracting sized blocks of adequate volumes. Fabric-related anisotropy is a key control of rock mechanical behavior under different environmental conditions, and the interplay between the fabric of crystal deformation and its brittle mechanical behavior needs to be taken into account properly throughout laboratory testing.

All these aspects require a multidisciplinary approach aimed at improving the resource efficiency and sustainability by:

1.  Using advanced technology for a detailed geologic, geomorphologic, and tectonic mapping for the validation of the deducted stress patterns.
2.  In-quarry aid to direct excavation to areas of the required quality.
3.  Assessment of the marble characteristics to obtain marketable products for specific uses.

While the geological and tectonic mapping provide qualitative analyses on large-scale areas (extraction basin and quarry scales), in situ stress state measurements give quantitative values of the acting stress state tensor. However, the two approaches are not standalone but have to be coupled for providing a comprehensive interpretation of the stress state.

This article presents a series of studies conducted by the authors over the last 20 years to make the exploitation of marble blocks in the Carrara basin safer, more efficient, and, therefore, more sustainable. It mainly aims to develop methodological and technological support for the mining sector covering both the theoretical and more practical aspects of natural stone exploitation.

The authors carried out studies in several fields, which are summarized below in this document:

*   Studies for the improvement of the knowledge of the natural stress state and stress induced by excavations through on-site measurements and numerical modeling [4–8].
*   Studies to improve the quality of the exploited material through improvements in cutting technologies [9–18].
*   Studies to improve the knowledge of the material's mechanical behavior under varying load and temperature conditions and to improve the understanding of the behavior of the material used for ventilated facades [7,19–23].
*   Studies to improve the reuse of waste materials and their reduction [24–26].

The present study is organized into six sections: the first one is introductory and presents the study area in which most of the presented studies were conducted. The second section provides information related to recent developments in the characterization of fracturing state and natural stress state. In "Technology improvements", the most recent extraction techniques used in marble exploitation are summarized. The section, "Mechanical behavior of marble in different environmental conditions", presents the results of recent studies performed on marble for evaluating its mechanical properties by considering temperature and chemical weathering as degradation factors. In "Extractive waste management and recovery", the methodologies and strategies used for a sustainable and efficient management and recovery of the extractive waste are described. Finally, the main findings of this paper are summarized and discussed.

## 2. Geological Setting of the Carrara Marble Basin

The Carrara marble basin (Figure 1) is located in the northernmost part of Tuscany (Italy) near the municipality of Carrara within the Apuane Alps (Northern Apennines). It has an extension of about 375 km$^2$ and is subdivided into three sub-basins—Torano, Fantiscritti, and Colonnata—from NW to SW.

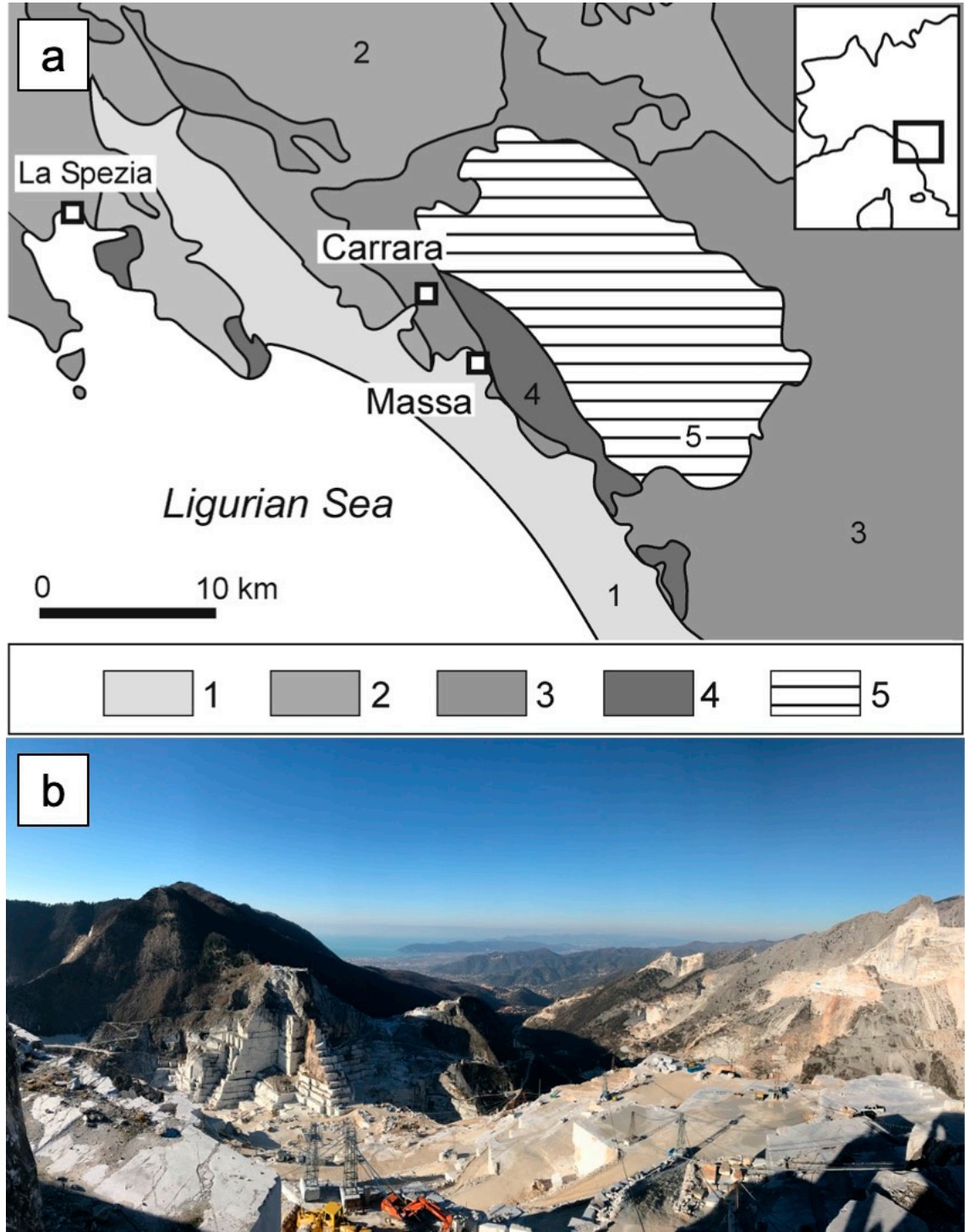

**Figure 1.** (**a**) Tectonic sketch-map of the Northern Apennines in the Carrara area. 1: Pliocene and Quaternary deposits; 2: Ligurian Units; 3: Tuscan Units; 4: Massa Unit; 5: Apuane Unit ("Autoch-tonous" auct.). Simplified from [27]. (**b**) Partial view of the Carrara marble basin from the top of the "La Gioia" quarry.

From a geological point of view, the Apuane Alps are divided into four tectonic units [26]: the Liguride and sub-Liguride systems, the Tuscan Nappe, the Massa Unit, and the Apuane Unit. The Liguride system is characterized by ophiolites, deep-water sediments, and sediments from the ocean–continental transition. Very low-grade and non-metamorphic sedimentary rocks characterize the Tuscan Nappe. The Massa Unit is made up of a Hercynian basement and a sedimentary cover,

deformed under higher metamorphic conditions. The same basement, unconformably covered by a sequence of sedimentary rocks metamorphosed in greenschist facies, can be recognized in the Apuane Unit.

The metamorphic Apuane Alps complex was generated by tectonic and metamorphic phases that occurred during the Alpine orogeny. The current structure was created by a sharp rise, initiated by an isostatic response to the previous doubling of the continental crust. Then, the Tuscan and Ligurian coverings were eroded, allowing the outcropping of a tectonic window through which the Apuane Alps emerged. The latter belongs to the geometrically lower Apuane Unit ("Autochtonous" Auct.), which crops out over a NW-SE trending elongated area ca. $20 \times 10$ km wide, and the overlying Massa Unit, which occurs as a relatively narrow, elongated belt at the inner border of the Apuane Unit (Figure 1a).

The well-known Carrara Marble derives from the greenschist facies metamorphism of the carbonate Liassic platform deposits (Marmi s.s. Formation) characterized by peak temperatures of 350–450 °C at pressures of 0.4–0.8 GPa [28].

The Apuan Unit's structural evolution includes two distinct tectonic-metamorphic events: the main event (D1), where the major structuration takes place within the metamorphic units. and the late structuration (D2), in which the metamorphic complex was gradually exhumed towards more and more superficial structural levels. The most recent stages of the D2 deformation are connected to a polyphase deformation that caused the development of brittle structures (fault and fracture systems) [29,30].

The Carrara marble has been exploited since Roman times for the maintenance of cultural heritage, art, and new building purposes, and is currently widely used in building applications all over the world.

## 3. Recent Developments in the Characterization of the Fracturing and Natural Stress States

The fracture characterization and the evaluation of natural and induced stress states and their relationships with extraction operations become essential to achieve the sustainability and competitiveness of stone extraction.

The state of stress in a volume of rock in the ground is determined by both the current loading conditions in the rock mass and the stress path defined by its geologic history. Changes in the stress state in a rock mass may be related to temperature changes; thermal stress; and chemical and physicochemical processes such as leaching, precipitation, and constituent mineral recrystallization. Mechanical processes, such as fracture generation, slip on fracture surfaces, and viscoplastic flow throughout the medium, can be expected to produce both complex and heterogeneous states of stress [31].

The need for reliable estimates of the pre-mining state of stress has promoted the development of stress measurement devices and procedures. Most methods use a borehole to gain access to the measurement site. The instruments for the direct and indirect determination of in situ stresses include photoelastic gauges, U.S. Bureau of Mines (USBM) borehole deformation gauges, biaxial and triaxial strain cells, flatjacks, and hydraulic fracturing. An alternative method for in situ stress estimation is based on the Kaiser effect and involves acoustic emission measurement [32].

The study of the geological structures can provide information regarding the regional tectonic events. In fact, the local stress field derives from the far-field tectonic stress, also known as the "far-field boundary conditions" [33], and it is an essential parameter for the definition of the geomechanical model. Unfortunately, it is not possible to directly measure past tectonic stress. Still, they can be reconstructed by analyzing the regional deformation structures in relation to the tectonic evolution of the area (paleostress) where the regional stress fields changed over space and time.

Focusing on underground structures, openings, and excavations (e.g., tunnels, quarries, mines, walls, etc.), their design and stability are strictly influenced by the local and regional stress fields in addition to rock mass properties [34]. The natural in situ stresses control the distribution and magnitude of the tensions around underground openings. Stress concentrations in the excavation

walls may be large enough to overstress the rock, exceeding the rock mass strength locally or at a large scale and thus inducing failure.

Methods of paleostress analysis from fault data (particularly fault-slip data) have been successfully used over the past five decades. Faulting is the primary indicator of stress variation; the regional tectonic stress strongly influences faults and fractures, and this is the reason why most studies that aim to characterize the stress system analyze the orientation, interaction, and fracture mode (opening, closing, shearing) of faults [35–38].

In addition, morphology and local geological structures can modify the orientation of the principal stress tensors and have to be considered during both the planning stage and the monitoring activities. The presence of faults or folds (even if they are generated by the regional stress state) can locally disrupt the regional stress field by changing its orientation, and fracture sets connected to those local structures may result in heterogeneous orientations, densities, and fracture modes [33,39].

In structural geology, a static or dynamic treatment of tectonic evolution is often introduced, considering the mechanical forces acting on limited three-dimensional regions that contain the analyzed structure. Therefore, geological and geomechanical surveys are fundamental steps in the work planning, particularly in the case of underground excavations, where both discontinuities and natural stress influence the design requirements and the induced stresses affect the void stability. To this end, a study of the regional tectonic setting and local structural features is fundamental, combining far-field tectonic stress, local geological structures, and morphology. These elements are necessary for correctly modeling the induced stress during excavation activities, planning in situ measurements, and monitoring stress variations.

Another major problem is related to the fracturing state of a rock mass; a careful study is necessary, since small variations in the degree of jointing could have negative consequences for rock exploitation.

This factor is the most important, since it controls a massif's ability to render blocks of suitable dimensions for processing, as there is a geometric limit below which the blocks cannot be commercialized [40].

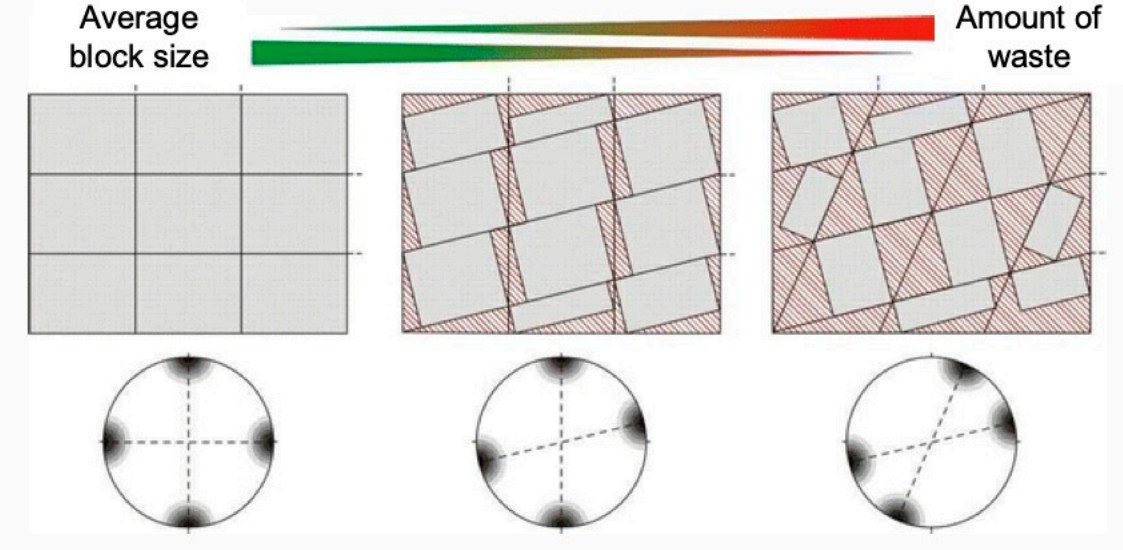

**Figure 2.** The amount of waste increases while the average extractable block size decreases due to the joint system deviating from orthogonality where the spacing pattern remains equal. The characteristic joint density distributions depict the orientations of each joint pattern (modified from [41]).

The genesis of joint and fracture systems can be multifaceted and traced back to orogenic and epeirogenic processes as well as shrinkage caused by cooling or desiccation. As discontinuities in a deposit, all surfaces such as faults, joints, cracks, fissures, or bedding planes must be considered [41,42].

The formation of individual blocks in a compact rock—the so-called in situ blocks [43]—is linked to the intersection of these discontinuities.

The model in Figure 2 shows in an exemplary way how changes in a system of orthogonal discontinuities occur when accompanied by rotation in one or both joint system groups. A comparison of the joint spacing distribution leads to the same results in all three cases. Furthermore, an increase in material loss occurs and a simultaneous decrease in the average block size can be documented [41].

In addition, the actual fracturing state of a rock mass could be a combination of the induced fracturing effect on the natural fracture system. If the extraction method causes significant fracturing, alternative methods should be considered [44].

Fracture mapping is the fundamental method for forecasting the number of marketable dimension stones [45]. Over the past two decades, fracture trace mapping evolved from 2D manual to 3D automatic or semi-automatic methods based on digital models [46–48]. These methods are particularly effective when the point density of the digital model is so high that its surface represents the actual shape of traces. Otherwise, boreholes images and Ground Penetrating Radar (GPR) or electrical resistance tomography could represent alternative methods for mapping fractures.

Discontinuity frequency (density) is one of the fundamental measures of the degree of fracturing of the rock mass. Frequency is used as a variable for geostatistical analysis; the fractal method has been applied for studying the fracture network [49,50]. Many studies focused on optimizing the exploitation planning based on multivariate geostatistical methods [51–56]. The results are used to generate a realistic 3D rock mass model which can be used in different applications, such as rock mass stability analysis, planning the exploitation system, or for sizing the exploitable blocks depending on the project goals [50,57].

The authors considered, among others, an underground quarry located in the Alpi Apuane marble district (Tuscany, Italy), named Ravaccione. Stress measurements were performed with a CSIRO cell in correspondence with horizontally drilled boreholes [4,7]. The results in terms of principal stress, magnitude, and orientation are reported in Table 1.

**Table 1.** Principal stress components with the mean orientation and magnitude measured in the Ravaccione quarry and the estimated orientation of the paleostress.

| Depth of Investigation [m] | Principal Stress | Magnitude [MPa] | Plunge [°] | Trend [°] | Plunge [°] | Trend [°] |
|---|---|---|---|---|---|---|
| | | **In-Situ Stress Measurements** | | | **Paleostress** | |
| 6.9 | $\sigma1$ | 16.5 | 79 | 242 | 78 | 47 |
| | $\sigma2$ | 1.3 | 10 | 086 | 10 | 263 |
| | $\sigma3$ | 0.5 | 4 | 355 | 6 | 171 |
| 9.65 | $\sigma1$ | 16.5 | 81 | 293 | 78 | 47 |
| | $\sigma2$ | 2.2 | 9 | 117 | 10 | 263 |
| | $\sigma3$ | 0.6 | 1 | 027 | 6 | 171 |

In addition, a multiscale and multidisciplinary approach was set up to detect and analyze geological structures to define the paleostress orientation. At the regional scale, a lineament extraction was performed employing a semi-automatic method [58] on the Digital Terrain Model of an area of about 10 km$^2$ in which the quarry is located (resolution of 10 × 10 m). The obtained lineament features (Figure 3a,b) were then associated with the main directions of the natural stress tensor (paleostress, Table 1) inferred from previous studies.

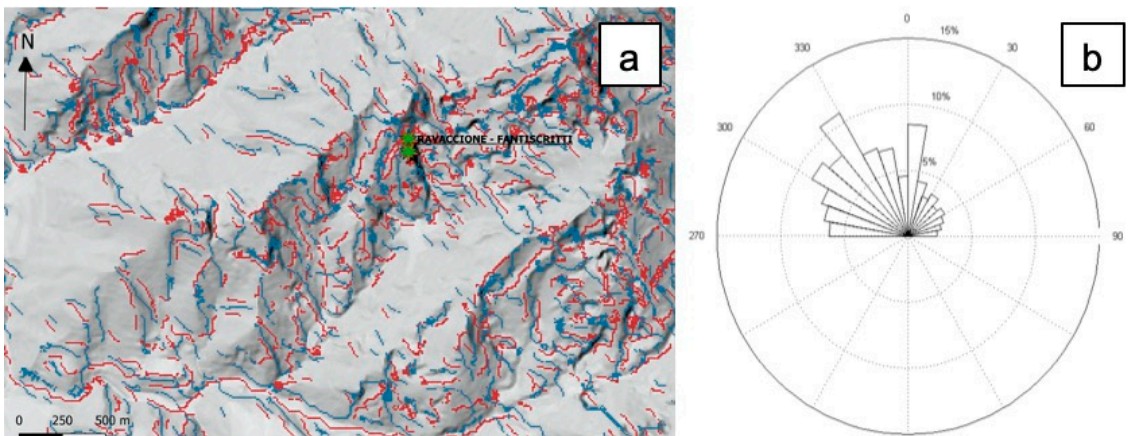

**Figure 3.** (**a**) Results of lineament extraction: map of the lineaments (convex in red, concave in blue), (**b**) rose diagram representing lineament frequency based on the direction with respect to north.

At the Ravaccione site (Figure 4), the quarry face is oriented nearly parallel and perpendicular to the maximum and minimum paleostress tensors. After this study, it was possible to define the variations in the local stress produced by the excavation at the time of the measurements: in particular, the vertical $\sigma_1$ stress changed in trend from about N47 to N293, while the horizontal $\sigma_2$ and $\sigma_3$ rotated about 180 degrees (Table 1).

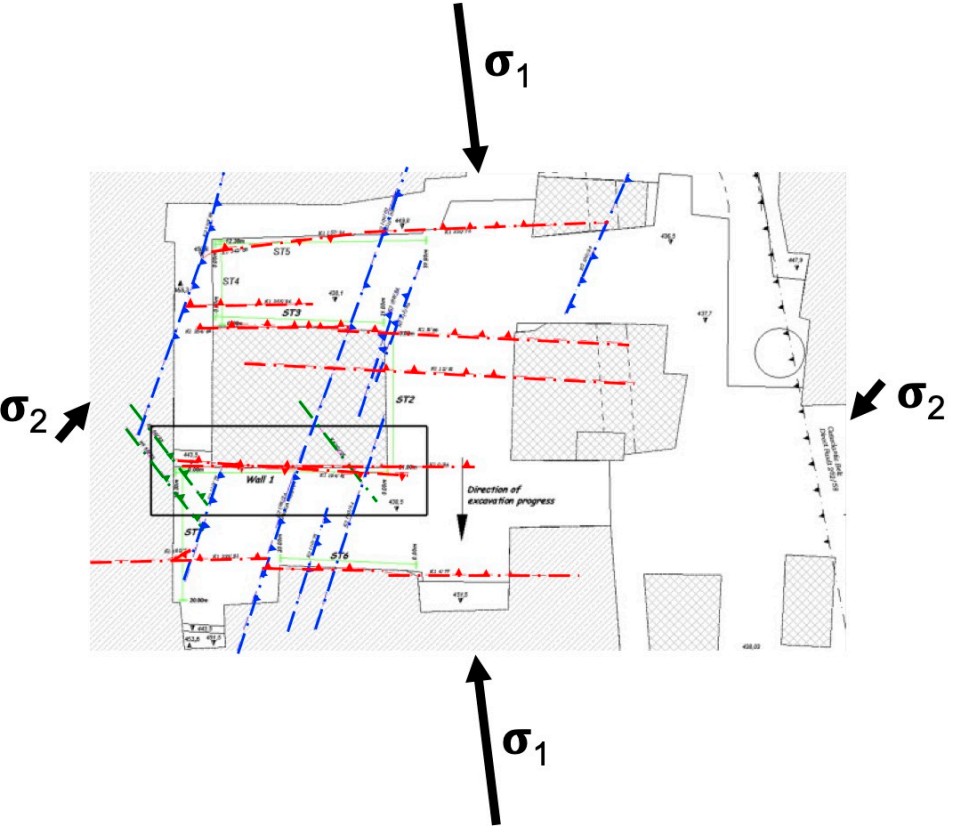

**Figure 4.** Map of the northwestern chamber of the Ravaccione quarry with the indication of the main surveyed discontinuities (red, blue, and dark green dashed lines) and the principal stress components measured using CSIRO cells ($\sigma_3$ is not reported because it is normal to the other stress components). The rectangular shape shows the location of the survey (modified after [7]).

## 4. Technology Improvements

The progressive optimization of extraction techniques and practices is essential to achieve a final sustainability and competitiveness of stone extraction.

In Italy, dimension stones represent a significant economic asset. The stone sector has progressively improved the available technologies, with the extensive use of shovels, front loaders, diamond wire saws, chain cutters, hydraulic drilling, and line drilling machines. Italian manufacturers mainly produce consumables and tools.

The technological evolution in the stone sector has led to a progressive increase in marble production in the Carrara basin, reducing excavation times and ensuring a better product quality. One of the most obvious consequences was the drastic reduction in the number of workers—this went from about 9000 to 800 in less than a century; on the other hand, the annual productivity/employee went from about 80 t to over 1000 t [59,60]. Furthermore, analyzing the period 1950–2020, a significant increase in productivity was recorded from 1976; this is due to the spread of diamond wire in the Apuan quarries. To date, over 20% of the material that has been historically excavated was exploited in the last 15 years. The annual production is around 1,400,000 t, representing the most important activity in the dimension stone sector; the density of quarries in that area is 0.33/km$^2$ (seven quarries/km$^2$ for the municipality of Carrara only) compared to the national average, which is 0.02/km$^2$ [59]. These results were achieved thanks to the high performance of the diamond wire cutters, which made it possible to adapt the cuts to the quarries' morphological conditions and guarantee a high productivity.

Diamond wire saws have greatly influenced both production speed and efficiency over the years. Despite the many improvements and optimizations over the years, their use is still the cause of injuries, sometimes fatal, following the splitting of the wire loops during cutting, resulting in whiplash and the high-speed projection of its components. Recent experimental studies have made it possible to evaluate some of the leading causes of the diamond wire loop splitting during cutting from some already known in the literature [61,62], such as failure due to the slippage of the connector crimp (Figure 5) and/or the breakage of the steel cable close to it, as well as other less known ones, such as the breakage of the connector crimps [63]. From these tests, it emerged that a knowledge of the materials used and a register showing the specifications of a given cut (i.e., length of the wire, cut surface, and so on) could lead to a more in-depth analysis of the causes of breakage and induce manufacturers to define diamond wire replacement criteria based on numerical values rather than on visual inspections dictated by the experience of the quarrymen, with the aim of safeguarding the health of workers and avoiding frequent breakage of the diamond wire.

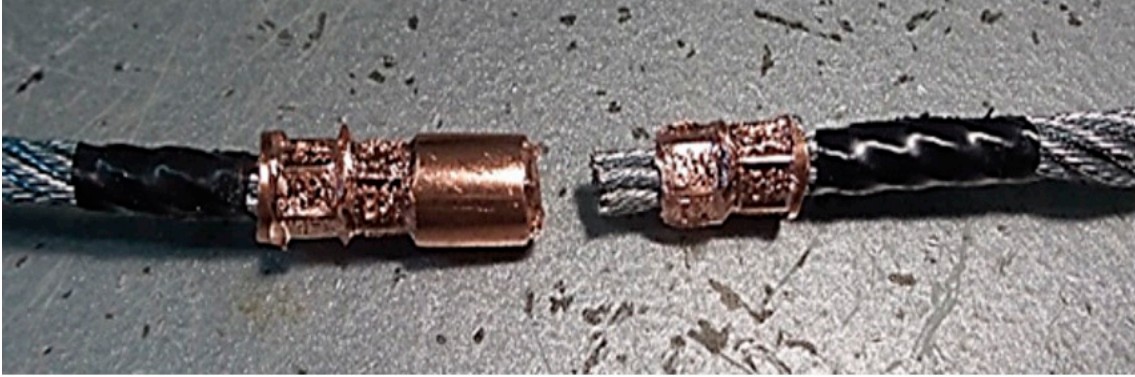

**Figure 5.** Breakage of the connector clamp, close of the variation in the resistant cross-section.

Another experimental campaign was recently carried out [64] to evaluate the actual load on the diamond wire during both the squaring and upstream cuts. The values provided by the measurements show stress on the wire of about 1/10 of its static tensile strength, considering a minimum value of 800 kg imposed by the connector crimp [65]. Therefore, if the cuts were made according to the

parameters specified during the acquisition, the wire's tearing should not occur unless a there is a sudden failure of the rock that could block the wire.

Today, the diamond wire cutter is a very versatile system and offers remarkable performance in the dimension stone cutting sector. The machines are equipped with safety systems which, however, can be improved: in this way, good working conditions could be obtained with respect to the cutting parameters identified during the experimental campaign (even in the case of upstream cutting (Figure 6), the load on the wire is lower than the limit imposed by the reference standards). This is why the need for technological evolution arises, and it is pursued by implementing the collaboration of electronics and control systems with the mechanics of cutting.

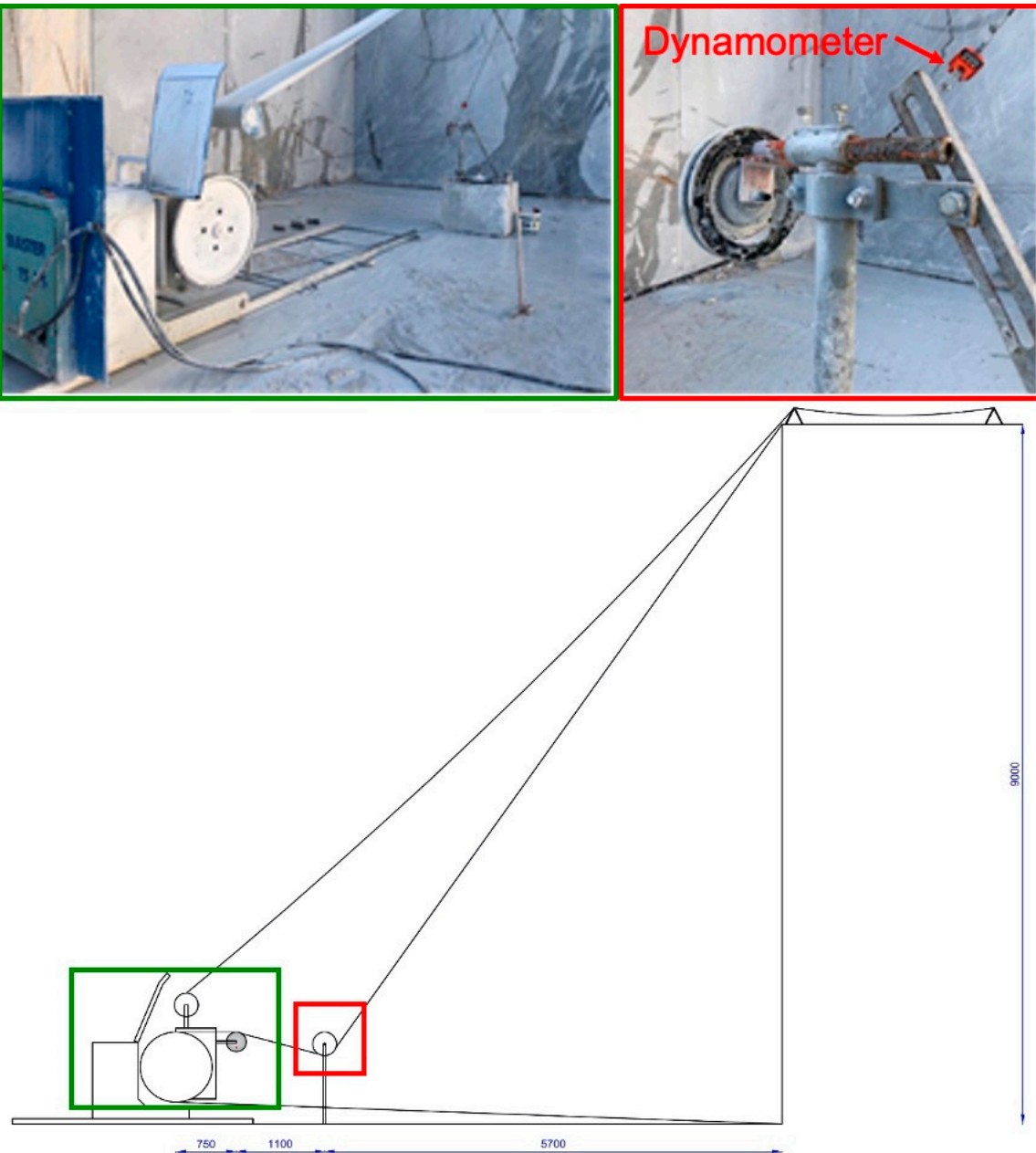

**Figure 6.** Cutting scheme through the Bfc MASTER TS75 cutting machine: upstream cut calibration, dynamometer on the wire's upper side.

One way to achieve this could be to install a bearing load cell on the machines, such as the one used for the experimental campaign (Figure 7), combined with a control system more refined than

those currently available. Thus, an automatic control system would be obtained that regulates the withdrawal based on the force acting on the wire without exceeding the load limit imposed by the standard and increasing the operation's degree of safety.

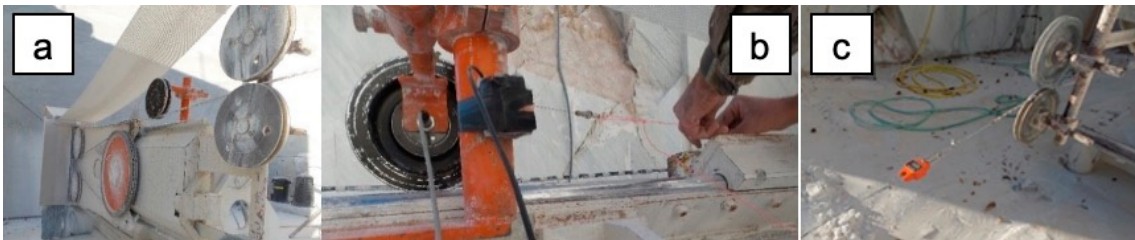

**Figure 7.** (**a**) Positioning of the load cell on the Apuania Corsi MF 5000 machine. (**b**) Placing the draw wire encoder. (**c**) Installing the dynamometer for calibration.

Using traction control, as required by UNI EN 15,163 [65], which limits the tension to the envisaged threshold, the probability of tool breakage would drastically decrease. However, this device would not prevent whiplash, which is a possibility since the electric motors cannot stop sufficiently quickly.

To do this, hydraulic motors controlled in feedback by servo valves would be necessary in order to obtain a system with a high dynamic response.

Another machine frequently used in marble cutting operations is the chain cutting saw, which has the great advantage of not requiring preliminary cutting operations. The arm operates easily, even in the presence of only one free surface. In all cases where it is necessary to perform blind cuts (as in underground excavation sites—Figure 8), the chain saw, alone or in cooperation with the diamond wire saw (Figure 9), is frequently used. Its limit is the length of the arm, which affects the depth of the cut, but recently very long arms (about 8 m) have been tested, although this may affect the cut's precision.

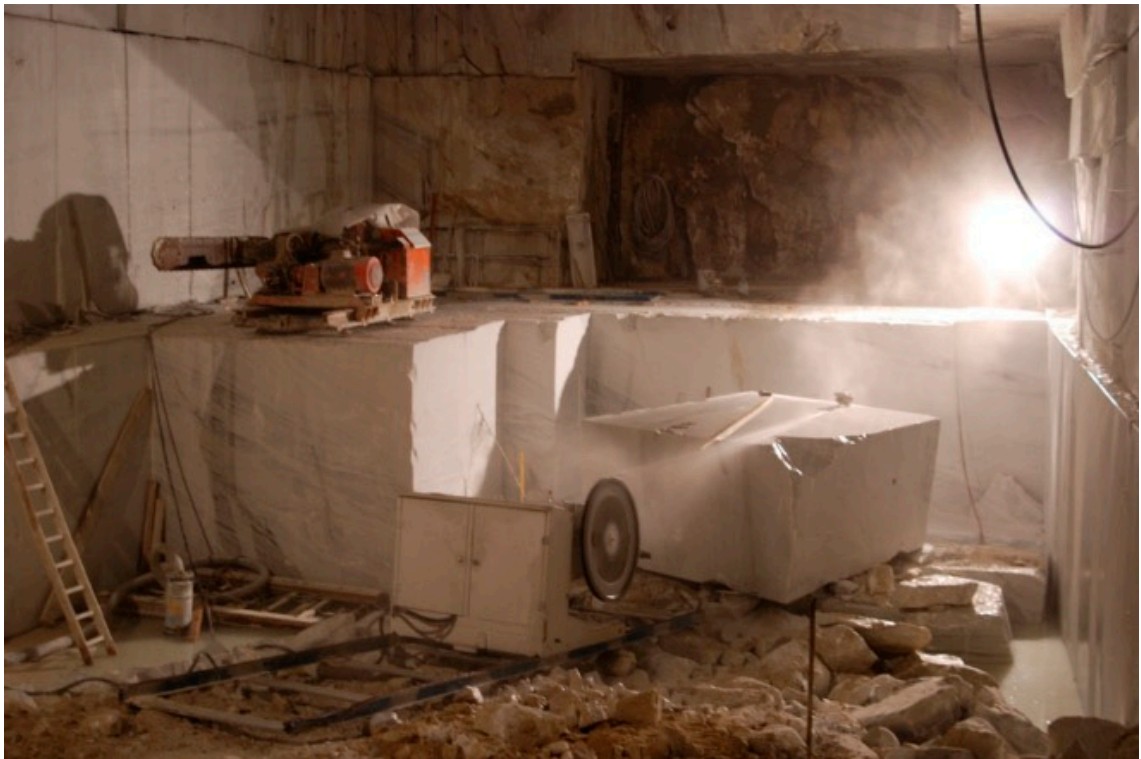

**Figure 8.** Chain cutting machine (on the top of the bench) operates in an underground extraction site (Lasa quarry), while a diamond wire saw performs a squaring operation.

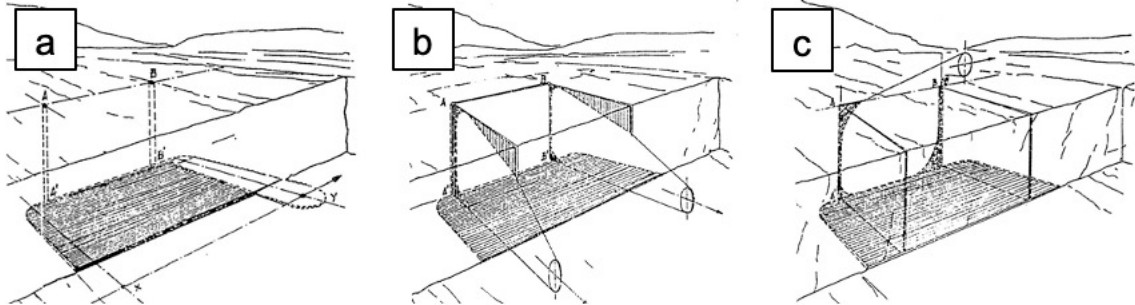

**Figure 9.** Cooperation between a diamond wire cutting machine and chain cutting machine. (**a**) Drilling of the vertical holes for the loop closure of the diamond wire. (**b**) Horizontal cut at the bottom of the bench through a chain saw (thickness of the cut: about 37 mm, enough to allow the passage of the wire circuit). (**c**) Cutting of the side surfaces and the rear surface using a diamond wire cutter.

In this field, studies are underway to improve the performance of the tools: the complexity of the cutting unit, which includes dozens of devices mounted at different angles, means that the inspection and replacement times are hours instead of minutes, and the environment in which research necessarily develops is a site much less equipped and comfortable than a mechanical workshop.

If the survey is limited to marble (homogeneous materials, at least for the purposes of cutting with this technique), it can be characterized by a strength value (uniaxial compressive strength) and an average value of hardness at small scale (HK, Knoop Hardness and/or HW, Wickers Hardness); tool materials should be tested separately for both hardness and toughness. The attack angle can only be changed by modifying the tool holder.

According to tests recently carried out in the laboratory [13,60], the bevelling of tools has improved their performance in terms of duration; this fact indirectly proves that small variations in the angle of attack can lead to significant improvements or detriments. Bevelling is equivalent to a local reduction, on a sub-millimeter scale, of the angle of attack; the operator decides on the depth of cut. Indeed, for a given series of tools, it depends on the length of the sequence (parameters with the operator cannot infer), the sliding speed of the chain, and the feed speed (progression of the machine): the operator adjusts the depth of cut through the speed of the machine, relying mostly on noise. A less subjective criterion would be desirable. Indeed, the operator will tend to increase the speed until the machine begins to vibrate abnormally. Still, the optimal progression speed (which ensures both good cutting speed and useful life of the tools and the machine as a whole) is certainly lower. Further investigations are underway, but it is believed that improved tool performance may lead to greater efficiency of quarry operations.

## 5. Mechanical Behavior of Marble in Different Environmental Conditions

In order to enlarge the market and the value of marble products, it is crucial to know how it will behave in different environmental conditions. The physical and mechanical properties of building stones can vary due to various degradation mechanisms caused by temperature and chemical agents. The problem of chemical and thermal weathering on marble rocks is an important issue to consider for designing building façades since it may cause sugaring, bowing, cracking, and spalling.

In a circular economy, a better awareness of the mechanical and physical processes to which marble products may be subject could increase the sustainability of exploitation processes by reducing the waste and the demand for replacing any products found to be faulty (e.g., marble slabs affected by bowing).

For this purpose, the authors conducted many studies for evaluating:

- The effects of temperature on the physical and mechanical characteristics.
- The combined effect of the thermal and chemical weathering.
- The effect of bowing on the marble slabs.

The effect of high temperatures as a degrading factor of rock materials has been investigated by [19,66]; marble samples were subjected to thermal cycles (ranging from 105 to 600 °C) and to subsequent non-destructive and destructive laboratory tests to evaluate the variation in the physical and mechanical properties as a function of thermal history after excavation (Figure 10). The study showed that the increase in crack density with temperature and the consequent porosity increases were the leading causes of degradation of physical and mechanical properties. In general, density, ultrasonic pulse velocity, wet electrical resistivity, uniaxial compressive strength, and Young's moduli decrease as temperature increases. By contrast, peak strain and porosity increase. Correlations between the temperature and physical-mechanical properties were proposed together with a damage parameter to quantify the degradation of mechanical properties as a function of the thermal history after excavation.

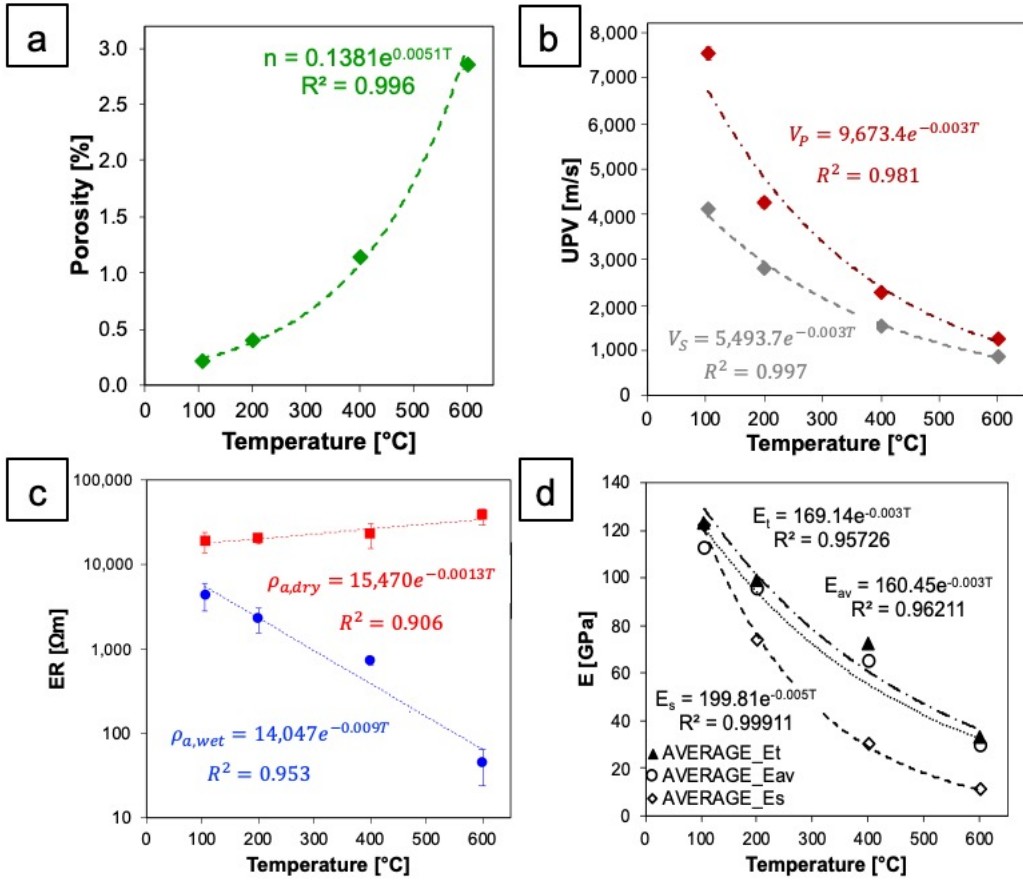

**Figure 10.** Relationship between (**a**) the average porosity, (**b**) the average P- and S-wave velocity, (**c**) the average dry and wet average electrical resistivities, and (**d**) the average values of Young's moduli of marble samples and the temperature to which the samples had been exposed.

In another study, the authors performed non-destructive (ultrasonic pulse velocities) and destructive tests (bending tests) on Carrara marble slabs in natural state and after thermal (with temperatures, respectively, of 50 and 90 °C) and thermo-chemical treatment. Thermo-chemical treatments were performed by soaking the specimens in a $10^{-6}$ mol/L solution of sulfuric acid at pH = 5 to simulate an acid rain exposure, at constant target temperatures, for one week. In general, for each weathering mechanism, the progressive degradation of the physical and mechanical properties of marble specimens was observed. In particular, a marked drop in tensile strength, mirrored by a wide variation in the P- and S-wave velocity, was found in specimens chemically treated at a target temperature equal to 90 °C (Figure 11).

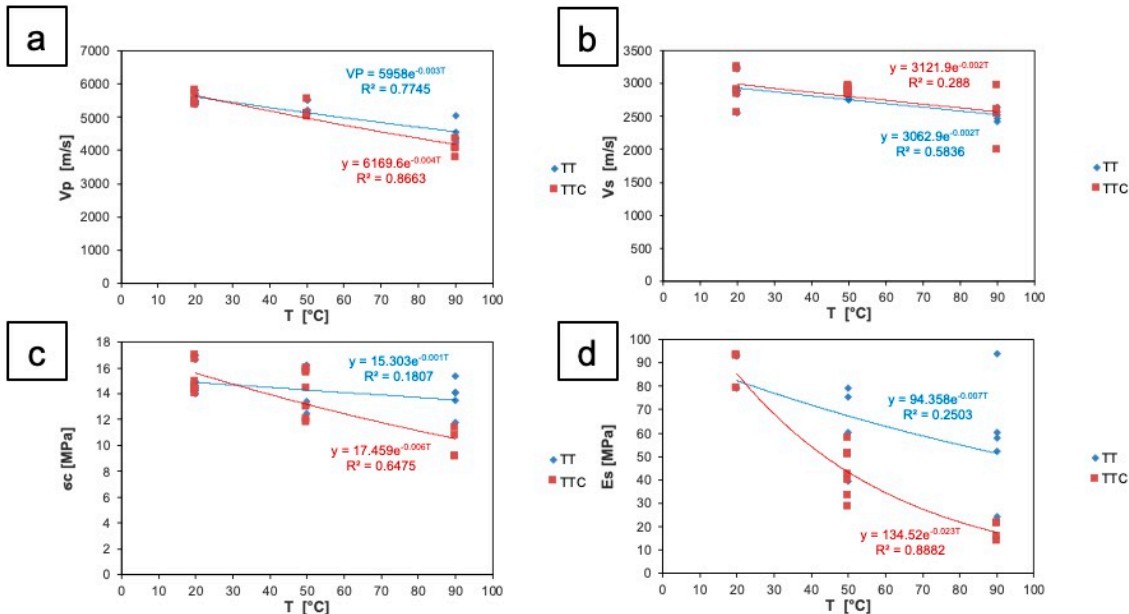

**Figure 11.** Relationship between (**a**) P-wave velocity, (**b**) S-wave velocity, (**c**) Shear resistance, and (**d**) secant Young's moduli as a function of temperature for Carrara marble sample after thermal (TT) and coupled thermo-chemical treatment (TTC).

The influence of marble degradation is particularly essential when marble slabs are used in ventilated facades (suspended facades), as they cover a large share of the ornamental stone market. The authors have focused on:

- Marble slabs in different environmental conditions by developing new models for forecasting the slabs behavior over time and proposing new technologies to improve the strength and durability of the slabs while preserving the aesthetic features of the natural material by using different anchoring systems. New products that can be safer, longer lasting, and of lesser weight can be developed, e.g., by impregnating the full thickness of the slabs with resins or designing composite slabs with backing layer of light material (honeycomb or foam).
- Optimization of tests for the certification of materials and improving predictive tools for structural design and life cycle analysis, to reduce the difference in terms of quality and reliability between natural and artificial materials.
- Improvement of the durability of the slabs with the aim of expanding the ornamental stone market by including applications with more severe environmental conditions (extreme temperature excursions and moisture content in Nordic, hot-humid, and desert environments; erosion by wind-transported particles; resistance to freeze-thaw cycles).

Figure 12 shows a scheme of the intergranular equivalent cracking considered for computing the bowing effects as a deflection of the slabs as a function of the months of exposition of a marble slab with a different grain density (Figure 13). The exposure to thermal cycles generates a diffuse cracking, mainly developing at the calcite grain boundaries, which can be schematized by means of crack density of the marble slab and incorporated in the present model to calculate the consequent convex bowing. The crack density parameter, n, that is the number of equivalent external edge cracks, can be correlated with the specific surface $S_s$ of the grains (i.e., the total surface area of the grains per unit volume of material) for a slab of length L, as reported in Figure 12.

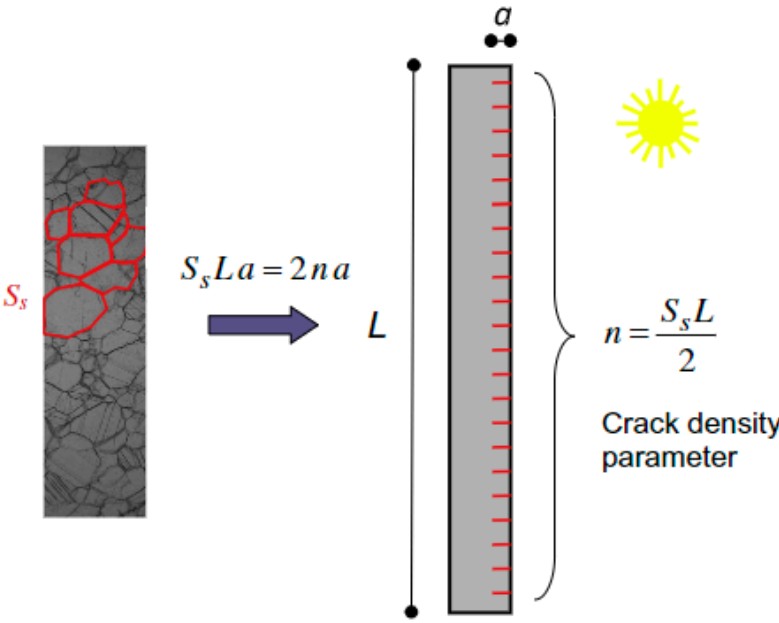

**Figure 12.** Scheme of the intergranular equivalent cracking (after [20]).

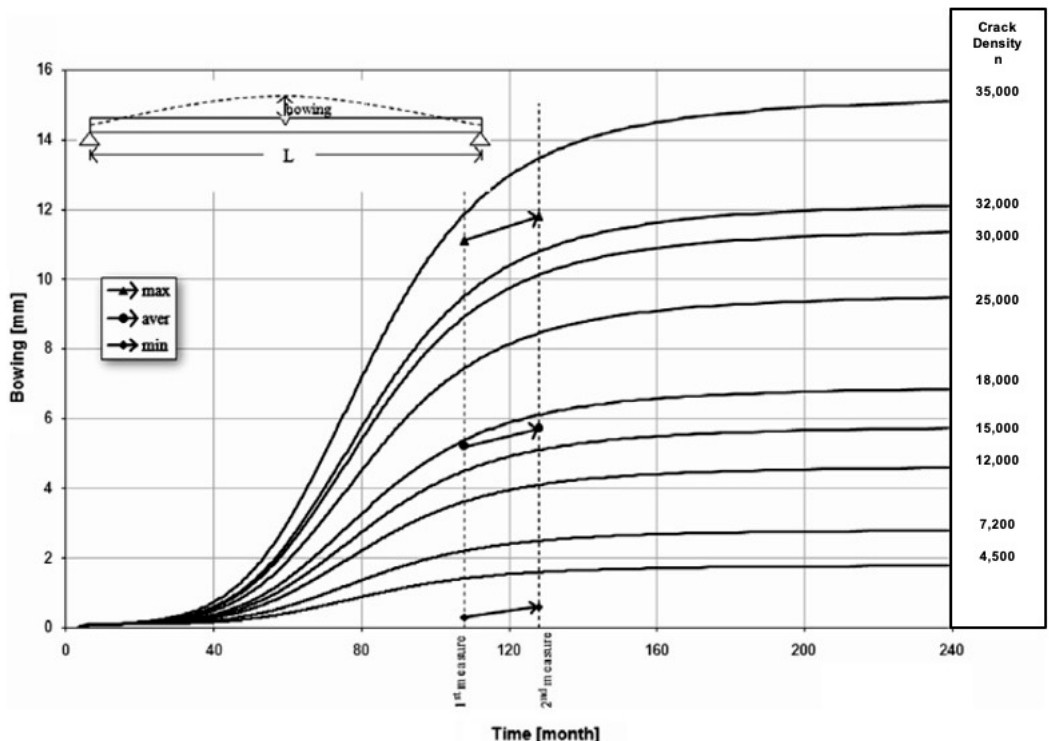

**Figure 13.** Total (bulk and crack-induced) deflection as a function of the number (expressed in months) of thermal cycles for the simply supported slab with different values of crack density n (after [20]).

## 6. Extractive Waste Management and Recovery

In 2016, the mining and quarrying industry represented the second most important sector, at the EU-27 level, in terms of waste production (27.6% or 624 million tons), just after Construction and Demolition Waste (C&DW—34.8% [67]). The need to minimize waste generation, in order to reduce its impacts on the environment and to conserve natural resources, and, at the same time, create the opportunity for the reuse/recycling of waste materials, is in line with the EU policy expressed in the Europe 2020 strategy for smart, sustainable, and inclusive growth [68] in the EU Sustainable

Development Strategy [69,70] and the Paris Agreement document [71]. Indeed, to strive for sustainable and efficient management and recovery of the extractive waste (EW), it is fundamental to guarantee the reduction in the environmental impacts associated with EW management. Moreover, it should be mandatory to ensure market conditions suitable for the new "recycled" products (by-products, secondary raw materials—SRM) coming from EW exploitation, together with higher awareness about the importance and convenience of using recycled products as alternative (integrative) to the ones coming from the exploitation of natural resources. The environmentally sustainable management of EW, which aims at recovering and recycling both clean and contaminated materials, would therefore help to reduce the pressure on natural resources and potentially to reduce the land use and the environmental and landscape pollution [25]. Territorial environmental agencies from Northern Europe recommend [72] the reuse of recycled products and SRM coming from waste treatment, whenever possible, instead of using natural (non-renewable) resources. In general, sustainable land and resources management is based on the overall organization of materials flows and on the optimization of the recycling activities, including the evaluation of the more suitable and marketable recycled products obtained from EW treatment and recycling activities [73].

At the Italian level, EW management, mainly associated with the dimension stone industry, is still a matter of concern, both for wastes produced in the quarrying areas and for the ones connected to working phases (cutting and grinding sludge, polishing, above all). Such wastes could be profitable and sustainable recovered and recycled as SRM (e.g., industrial minerals, high-value products, aggregates, filler materials, etc.). Still, most of the time, they were disposed of in EW facilities. The Carrara marble quarrying basin represents a vital case study for the recovery and recycling of EW. Indeed, EW recycling, together with more modern and efficient quarrying techniques and technologies, leads to the aimed for sustainable mining of the marble resources.

The Carrara quarry basin includes about one hundred quarries for colored and white marble exploitation, exploited since Roman times. The quarry basin is split into three sub-basins (NW to SW: Torano, Fantiscritti, and Colonnata sub-basins). Marble was exploited with traditional systems up to the 1990s; marble production was originally for national and international customers, characterized by a low-medium production rate and a "not-industrial" approach. Such a kind of quarry management allowed the coexistence of the *Ravaneti* (EW facilities; some historical *Ravaneti* are still visible and renewed) and quarries as natural parts of the landscape, guaranteeing the safety of slope stability and a "green" environmental rehabilitation. From the 1990s, there was a huge increment of marble demands from foreign countries (China, Russia, and Emirates above all), causing a fast and intensive quarrying growth. The EW volume increased and increased (Figure 14). The blocks' size differed depending on the different working era, quarrying techniques, and technologies applied (e.g., modern cutting systems generate more fine materials).

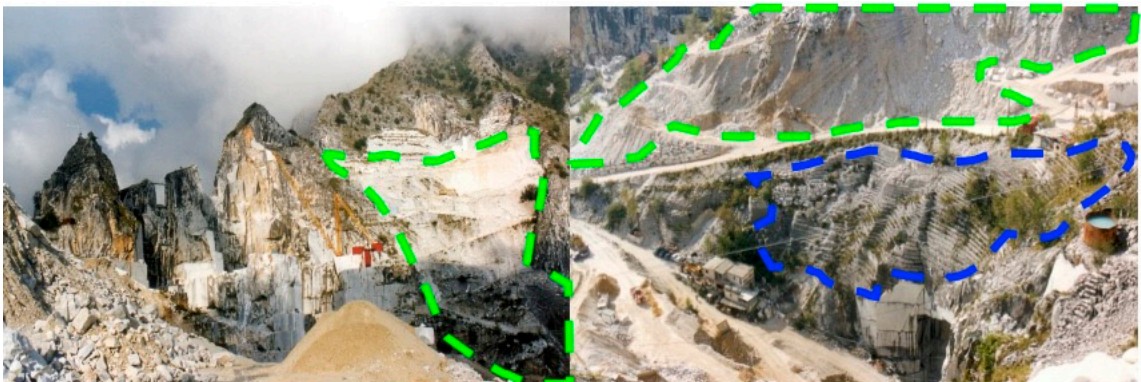

**Figure 14.** Carrara marble quarry basin: open pits and *Ravaneti* (green dashed lines). Some of the historical *Ravaneti* (blue dashed line) have been used as a base for the internal roads in quarrying areas.

At present, the EW generation is about 3 $Mm^3$/y (fluent waste); another 80 $Mm^3$ of EW still present in the *Ravaneti* has to be added to the fluent wastes, and only 0.5 $Mm^3$/y of EW is exploited for SRM production [25]. The amount of EW has become larger and larger, generating problems in quarry management and activities (landslides, the flooding of roads and inhabited urban areas). In the meantime, leading companies in calcium carbonate production built dressing plants to recycle material present in the *Ravaneti*, from EW to valuable industrial products, recovering only the pure under-sized calcium carbonate coming from quarries and leaving a great part of the EW (the worst in quality) in the quarries. In the 2010s, a Newco (Carrara Marble Way srl, which groups 40 Carrara and Massa quarry company) decided to invest in EW management and recovery, creating a complete production line to exploit EW (from the best to the poorest quality). The starting point to evaluate the best potential applications for marble waste (both coarse and fine fractions) was to investigate different research dealing with the recovery of limestone and marble waste [74–79], including also the residual sludge produced during drilling activity and in processing plants [26,80–84].

Carrara marble characteristics are reported in Table 2, while Table 3 summarizes specific tests on two different EW typologies present in the Carrara quarrying basin (Calocara and Lorano quarrying areas). For each area (Lorano and Calocara), tests have been performed for three different size classes: 0.5–4, 0–25, and 0–150 mm.

**Table 2.** Average Carrara marble physical-mechanical characteristics.

| Physical Characteristics | Value |
|---|---|
| Bulk density | 2688 $kg/m^3$ |
| Simple compression strength | 1209 $kg/cm^3$ |
| Compression strength after freezing | 1181 $kg/cm^3$ |
| Indirect Tensile Strength (Brazilian test) | 174 $kg/cm^3$ |
| Impact strength test | 73.8 cm |
| Moisture absorption (by weight) | 0.16% |

**Table 3.** Summary table of the results of the physical tests [25].

| Sample Name | Grain Size Distribution | Atterberg Limits Liquid Limit WL% | Atterberg Limits Plastic Limit WP% | Density (Average) | Los Angeles Test % | Freezing and Heat Test (Average) % | Shape Index (%) | Flatness Index (%) |
|---|---|---|---|---|---|---|---|---|
| C 0.5–4 | Sand slightly gravelly | - | | 2.55 | | - | - | - |
| C 0–25 | Sandy gravel slightly silty | Not plastic | | 2.59 | 68 | 0.8 | 16.5 | 27.2 |
| C 0–150 | Gravel slightly sandy-silty | Not plastic | | 1.96 | 69 | 0.3 | 17.4 | 19.5 |
| L 0.5–4 | Sand slightly gravelly | - | | 2.46 | | - | - | - |
| L 0–25 | Sandy gravel slightly silty | Not plastic | | 2.40 | 43 | 0.4 | 21.9 | 25.7 |
| L 0–150 | Gravel slightly sandy-silty | Not plastic | | 1.98 | 42 | 0.3 | 28.8 | 29.5 |

The first results (grain size distribution, Los Angeles test, freezing and heat test, and Atterberg limits, Table 3) show a possibility for EW to be recovered as crushed materials for embankments.

The sampled materials show the following geochemical characteristics: $SiO_2$ (0.32–2.63 wt%); $TiO_2$ (0.1–0.4 wt%); $Al_2O_3$ (0.04–0.9 wt%); $Fe_2O_3$ to (0.04–0.40 wt%); MnO (<0.1–0.2 wt%); MgO (0.72–2.13 wt%); CaO (52.60–56.10 wt%); $Na_2O$ (<0.01 wt%); $K_2O$ (<0.01–0.21 wt%); $P_2O_5$ (<0.01–0.04 wt%); $Cr_2O_3$ (<0.01 wt%); BaO (<0.01 wt%); SrO (0.01 wt%); LOI (42.00–43.00 wt%). From a petrographical point of view, Calocara is an extremely pure, relatively coarse-grained marble showing a granoblastic texture, while Lorano is an almost pure granoblastic marble which is, however, characterized by a variable grain size. The EW coarse fractions show the same characteristics as the original rock, while the EW fine fractions, compared to the primary rock type (which is a high-purity marble), are slightly enriched in quartz (Calocara: <1 vol. %; Lorano: ~1.5 vol. %), white mica

(Calocara: <1 vol. %; possibly partially chloritized; Lorano: ~2 vol. %), and Fe oxides/hydroxides (Calocara and Lorano: <0.5 vol.) [25].

Based on the geochemical, mineralogical, and petrographic characterization, both Calocara and Lorano EW (coarse and fine fractions), extremely pure in calcium carbonate, may represent high-value products that could be used as filler for the production of paper, rubber, paint, plastic, etc. [85].

The results obtained from EW characterization (carried out in two of the most important quarrying areas in the Carrara marble quarry basin) were the basis to design and construct a proper processing plant (selection, crushing, and screening) to exploit the valuable materials disposed in the *Ravaneti*. The treatment plant still treats the EW, even if, due to the recent global market crisis, it is harder to sell the produced SRM and, consequently, the amount of material fed to the processing plant is decreasing more and more.

In summary, at a broader scale, the extractive industry should aim to guarantee the systematic and profitable recovery of EW as SRM; to this end, a change in quarrying and working activities has to be projected. Together with a more efficient extraction, a dedicated disposal area for marble scraps is needed. Furthermore, several actions are fundamental to improve the efficiency of EW exploitation:

- The definition of a working protocol that indicates how to manage the EW and which characteristics are required for each new product (e.g., the crushed materials for embankments must be separated from the high-quality ones so as not to dilute their quality).
- Cooperation between public authorities and industries to define guidelines and operative protocols for the application of SRM at a broader level (e.g., in public works and infrastructure).
- A market ready to accept new products obtained from EW processing. To this end, it is necessary to inform and sensitize the civil society about the necessity of accepting and using products from "waste" processing (End of Waste Criteria). Indeed, the SRM obtained from EW facilities represents an important source, in addition to the RM coming from virgin deposits. Waste must be considered a future resource, and waste facilities have to be considered "new ore-bodies" to exploit following the "mining approach" principles (which deal with the actions that we need to evaluate if the exploitation of the resources present in EW facilities is feasible. Geopolitical, environmental, economic, and social contexts have to be analyzed, together with the characterization and estimation of the resources present in EW facilities [84]).

## 7. Conclusions

The sustainability of the extraction of natural stones for building purposes, such as tiles and slabs, requires optimizing all phases of the production process from exploitation to final use. The challenge is to produce goods that combine ecology and economy, ensuring high production standards that are compatible with the historical heritage buildings and architectural traditions in which they will be installed and minimizing the generation of waste.

The assessment of the characteristics and environmental impacts of natural building stones is the core of sustainable exploitation processes. To reach these goals, studies have been conducted by the authors of all the production steps in order to improve our knowledge of the state of stress and the rock mass fracturing determination, the thermo-chemo-mechanical behavior of rocks, the cutting techniques used, and the strategies for waste reduction.

This paper reported the latest advances in all these fields. In particular, non-contact techniques allow the estimation of the main lineaments that can influence the recovery rate and material quality. The studies concerning the cut stone behavior under different environmental conditions, in terms of temperature and chemical interaction, are at the base of ongoing research devoted to setting up processes able to produce slabs meeting the specified quality requirements, making natural products competitive with artificial ones. Waste reuse is a promising way to increase economic income by finding further employment in the industry.

The application of the described methodologies is expected to significantly contribute to the reduction in the cut stone environmental impacts in terms of: (i) a reduction in the stone wastes to

be disposed, (ii) the rational management of stone resources, and (iii) a reduction in the total energy consumption in exploitation phases.

Future research should be devoted to the recovery of each fraction produced during cut stone processes and storing the residual waste in mapped and known facilities in order to make it, on the one hand, easily monitored and, on the other hand, accessible for possible future exploitation.

**Author Contributions:** Conceptualization, A.M.F.; methodology, F.V., G.A.D., G.U., M.C., and A.M.F.; writing—original draft preparation, F.V., G.A.D., G.U., M.C., and A.M.F.; writing—review and editing, F.V., G.A.D., G.U., M.C., and A.M.F.; visualization, F.V. All authors have read and agreed to the published version of the manuscript.

**Funding:** This research received no external funding.

**Conflicts of Interest:** The authors declare no conflict of interest.

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
