# Peer review of "New Developments for the Sustainable Exploitation of Ornamental Stone in Carrara Basin"

_sustainability, doi:10.3390/su12229374_

Round 1

Reviewer 1 Report

I have made extensive comments in the enclosed version of the manuscript that are hopefully helpful.

Perhaps the introductory section could make stronger emphasis of the need for a co-ordinated approach to improve resource efficiency, involving geologic/tectonic mapping, geophysical methods to confirm the deducted stress patterns and in-quarry aid to direct excavation to areas of the required quality and assessment of the marbles to obtain marketable products for specific uses.

EW management and utilisation is both, an element of resource efficiency and a tool to improve 'sustainability' of the extractive industries.

You may also want to look at the results of the EC FP7 project SUSTAMINING that cover many of the aspects discussed in your paper.

Author Response

Dear Reviewer,

thank you for your precious comments and suggestions. Please see the attached file that includes the point-by-point reply to your remarks. 

Best regards

Reviewer 2 Report

Dear Authors,

Thank You for the opportunity of reading this article. My general opinion about the article is positive.

General statements:

-> article present studies aimed at making the exploitation of marble blocks in the Carrara basin safer, more efficient, and, therefore, more sustainable. The presented topic is suitable for the scope of the Sustainability journal.

-> the content of the article is sufficient to be published in a good journal.

-> abstract is adequate for the content of the article.

-> literature background is based on 81 literature positions. However, most of them are older than 5 years.

-> the organization of the article is well.

-> quality of figures and tables is high.

However, I indicated some elements that require revision:

#1 adding a reference

Add literature reference to statement indicated in lines 30-32.

#2 Introduction

-> Please consider the reorganization of the introduction. I think adding additional subsections would increase its legibility.

-> Please also add in the introduction a short description of the article organization. I mean “Section 2 presents …, section 3 concerns….. etc.”

-> Also please strongly highlight in the introduction the part with the contribution of this paper.

#3 dot

Please add a dot after the last keyword (line 27).

#4 Colors in text

Please change the figure and literature reference to black color (red and blue are not correct).

#5 extend conclusions

Please extend the conclusions by adding future research directions.

Author Response

(The authors gave the same response as above.)
